# Hydrodynamics and direction change of tumbling bacteria

**Mariia Dvoriashyna**[ID]**, Eric Lauga**[ID]*

Department of Applied Mathematics and Theoretical Physics, University of Cambridge, Cambridge, United Kingdom

* e.lauga@damtp.cam.ac.uk

**Data Availability Statement:** All relevant data are within the paper and its Supporting information files.

**Funding:** This project has received funding from the European Research Council under the

## Abstract

The bacterium *Escherichia coli* (*E. coli*) swims in viscous fluids by rotating several helical flagellar filaments, which are gathered in a bundle behind the cell during 'runs' wherein the cell moves steadily forward. In between runs, the cell undergoes quick 'tumble' events, during which at least one flagellum reverses its rotation direction and separates from the bundle, resulting in erratic motion in place and a random reorientation of the cell. Alternating between runs and tumbles allows cells to sample space by stochastically changing their propulsion direction after each tumble. The change of direction during a tumble is not uniformly distributed but is skewed towards smaller angles with an average of about 62˚–68˚, as first measured by Berg and Brown (1972). Here we develop a theoretical approach to model the angular distribution of swimming *E. coli* cells during tumbles. We first use past experimental imaging results to construct a kinematic description of the dynamics of the flagellar filaments during a tumble. We then employ low-Reynolds number hydrodynamics to compute the consequences of the kinematic model on the force and torque balance of the cell and to deduce the overall change in orientation. The results of our model are in good agreement with experimental observations. We find that the main change of direction occurs during the 'bundling' part of the process wherein, at the end of a tumble, the dispersed flagellar filaments are brought back together in the helical bundle, which we confirm using a simplified forced-sphere model.

## 1 Introduction

Many of us are surprised at first when we realise that bacteria represent the major component of the world's biomass [1]. A number of these bacteria are motile and swim with the use of flagellar filaments, which are slender helical appendages [2], enabling the cells to locate nutrients using chemotaxis, a process that is now well understood [3]. Most of what we know about chemotaxis was obtained using the model bacterium *Escherichia coli* (*E. coli*), which is peritrichous, meaning that it has several flagella distributed over the cell body [4].

The helical filament of a flagellum, typically O(5–10) $\mu$m in length and 40 nm in diameter, is attached to a cell body via a short flexible hook connected to a rotary motor embedded in the cell wall [3]. The motor rotates with an approximately constant torque [5], at a rate of

European Union's Horizon 2020 Research and
Innovation Programme (Grant No. 682754 to E.L.).

**Competing interests:** The authors have declared
that no competing interests exist.

about 100–200 Hz, resulting in a passive rotation of flagellar filaments, which allows the cell
body to be propelled forward [6]. Each helical flagellar filament is a polymer of a single protein
called flagellin [7]. Depending on the conformation state of this protein, each filament can
take different polymorphic forms, resulting in different possible curvatures and twists of the
helix. There are twelve polymorphic forms that are molecularly and mechanically stable, two
of which are straight and ten are true helices [8]. The polymorphic form used for straight
swimming by wild-type bacteria is the so-called 'normal' form [9], which was shown to be the
most hydrodynamically efficient one [10].

Fluid mechanics has proven to be an important tool to help understand the motion of small
organisms in fluids [11–14], such as protozoa [15], aquatic microorganisms [16–18], sperma-
tozoa [19, 20] and bacteria [9]. In most of these studies the viscous forces are dominant, as the
Reynolds number typically ranges between $10^{-4}$ and $10^{-2}$, thus placing cell swimming well in
the Stokes flow regime in which inertial effects can be neglected. An early study of bacterial
hydrodynamics appeared in the seminal work of Chwang and Wu [21]. They used the resis-
tive-force theory of Gray and Hancock [22], valid for slender filaments, and enforced the bal-
ance of force and moments on the cell, which allows one to compute simultaneously the linear
and angular velocity of the organism. Since then, there has been a lot of research in bacterial
hydrodynamics aimed at understanding the interactions of bacteria with surfaces [23–31],
with other cells [32] and their collective motion [33–36].

In order to sample the environment and change their swimming direction, flagellated bac-
teria have adapted various strategies, the most studied being the so-called 'run and tumble'
motion [9], used by peritrichous bacteria such as *E. coli*. During a run, the cell moves forward
(approximately steadily) and its filaments are gathered in a helical bundle behind the cell body,
which rotates in a counter-clockwise (CCW) fashion when viewed from behind the cell. In
between runs, the cell undergoes quick tumble events, during which at least one filament
reverses its rotation direction, changes its polymorphic form and separates from the bundle,
resulting in erratic motion in place [3, 37]. At the end of each tumble all motors return to their
CCW rotation and the bundle is reassembled (see sketch in Fig 3). Alternating between runs
and tumbles allows bacteria to sample space by stochastically changing their propulsion direc-
tion after each tumble [38], as in a random walk. The duration of runs is about 1 s, whereas
tumbles are approximately 0.1 s long. If a chemical attractant is present, the random walk of
the cell becomes biased towards the attractant and the tumbling frequency is reduced [39–41].

Statistically, the change of direction of the cell after a tumble (denoted here by the angle $\gamma$)
is not uniformly distributed. Instead, the probability density function is skewed towards
smaller angles and has an average of about 62˚–68˚, as shown in Fig 1 where we reproduce the
pioneering 1972 measurements from Berg and Brown [38]. The distribution of angle $\gamma$ was
measured in many other studies since then, with similar average values reported: 57˚±42˚ [37],
57˚±37˚ [42] (with the distribution also displayed in Fig 1) and 60˚±29.2˚ [43]. The value of
the mean angle during tumble is important because it controls the effective diffusion of the
cells; for a cell running with a mean speed $U$ and tumbling on average every time $T$ the diffu-
sion constant for the cell is given by $D_{\text{eff}} \sim U^2 T/(1 - \langle \cos \gamma \rangle)$ [44]. Note that the tumbling pro-
cess has been modelled as effective rotational diffusion in several works [45, 46] and, although
such approach captures well the statistics of tumbles, the detailed mechanism by which cells
change their directions remains elusive. The aim of the present work is to combine geometrical
and hydrodynamic modelling to reproduce the distributions shown in Fig 1.

At the heart of the tumble are two complex fluid-structure interaction processes: (i) 'unbun-
dling', when the filament leaves the bundle, and (ii) 'bundling', when the bundle reassembles
at the end of a tumble. Several studies have focused on the key role played by the instability of
the hook, which has a bending rigidity several orders of magnitude smaller than that of the

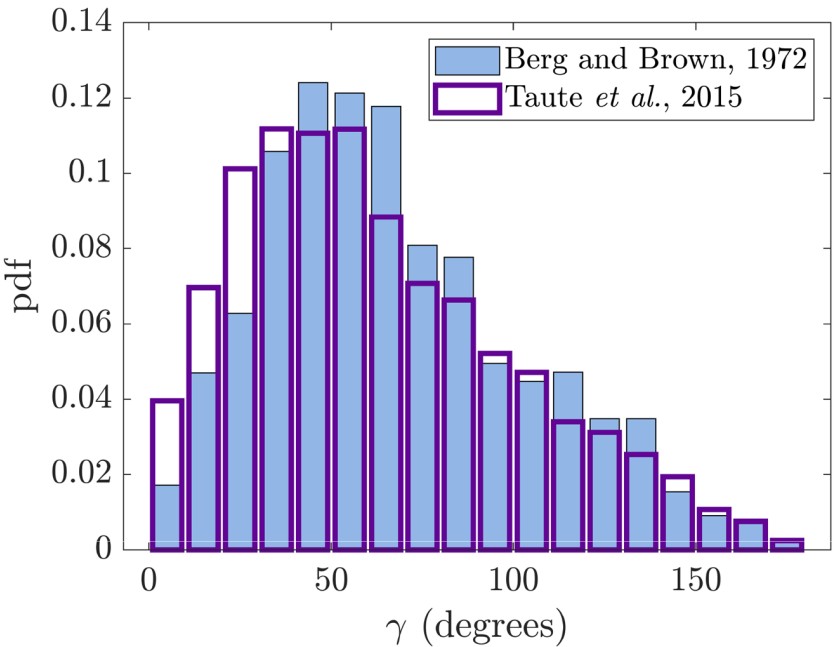

**Fig 1. Experimental probability density function (pdf) for the angle γ (in degrees) measuring the change in direction from the end of one run to the beginning of the next for wild type *E. coli* cells, reproduced from Berg and Brown [38] (blue) and Taute *et al*. [42] (hollow with purple edges).**

filament [47], in the bundling process [48] and thus in controlling the change in the swimming direction [49, 50]. The synchronisation between flagellar filaments during the bundling process is also an important factor. Numerical simulations have been used to study the synchronisation between rotated helices [51, 52], and so have macro-scale bundling [53] and flow field visualisation studies [54]. It was discovered that hydrodynamic interactions between helices are sufficient to induce attraction, wrapping and synchronisation of the filaments, if the helices were of the correct combination of chirality (both left-handed and rotating CCW or their mirror-image equivalent), although some elasticity was required—this is presumably provided in real cells by a combination of the hook and the flagellar filaments themselves. Recently, a theoretical work addressed the role of indirect hydrodynamic interactions between flagellar filaments in the bundling process [55]. Finally, a number of numerical studies have modelled the full bundling/unbundling dynamics, including the elasticity of the flagellum, using various computational methods, such as regularized flow singularities [56], mesoscle methods [57, 58], bead-spring models [59, 60] and boundary element methods [61, 62].

In this paper, we use a combination of geometrical and hydrodynamic models to predict the change in swimming direction occurring during a tumble, with the ultimate goal of reproducing the distribution in Fig 1 theoretically. We first use the experimental observations in Ref. [37] where a fluorescence imaging technique is developed for the visualisation of flagellar filaments during the swimming of the cells on which they are attached. This allows us to construct a kinematic model of the flagellar filaments during a tumble in Section 2. We first describe kinematically in Section 2.1 the helical interactions occurring during polymorphic transformations based on Refs. [63, 64] and model separately the unbundling and bundling processes. We then incorporate in Section 2.2 this geometrical description into a hydrodynamic model based on the resistive-force theory of slender filaments [65] and obtain the linear and angular velocities of a cell body during tumble. The results of the model, detailed in

Section 3, allow us to compute the change in swimming direction after a tumble. This in turn can be used to obtain the statistical distribution of tumble angles for a cell with several flagella distributed uniformly on the surface of its body, which we find to be in good agreement with Fig 1. Further analysis reveals that the bundling stage of the process is responsible for the majority of the cell reorientation during a tumble, which we confirm using a simplified forced-sphere model in Section 4. We summarise and discuss our results in Section 5, highlighting the role of geometry and mechanics in setting the reorientation angle.

## 2 Modelling of an individual tumbling event

In this first part of the paper, we present the details of our model, which combines geometry with hydrodynamics.

We model the cell as a spherical body of radius $a \approx 1$ $\mu$m equipped with $N$ rigid helical flagellar filaments (with $N$ typically between 2 and 6). The filaments are assumed to have contour length $L \approx 10$ $\mu$m and thickness $2h \approx 40$ nm [3]. The organism is immersed in a viscous fluid, with dynamic viscosity $\mu$, which we assume to be that of water, $\mu \approx 10^{-3}$ Pa · s. When the flagellar filaments are in the bundle, we assume that they are all aligned with one axis (this is the $z$-axis in the body frame in the first two stages of the tumble, see notation in Fig 2). A filament outside the bundle is located at an azimuthal angle $\varphi$ and a polar angle $\theta$ relative to the axis of the bundle.

In the model, we choose the axes of all flagella to be aligned with an outward normal to the spherical body. In experiments, the filaments do not always remain normal to the cell body, and instead their direction is governed by the balance between hydrodynamic stresses and the elastic resistance of the hook, see e.g. Ref. [50]. To keep the model as simple as possible, we assume here that the filaments remain normal to the cell body, but that they are allowed to

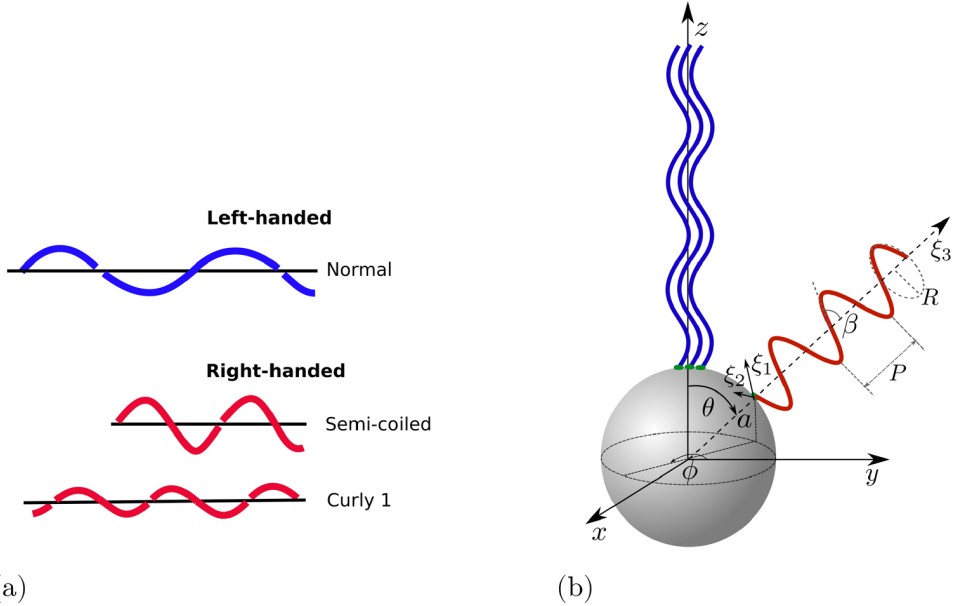

(a)                                                                              (b)

**Fig 2.** (a) Schematic representation of flagellar filaments with three different polymorphic forms: normal, semi-coiled and curly 1. (b) Notation for the model bacterium and the coordinate system used in the body frame. Filaments in blue are in the bundle and are aligned with $z$-axis. The filament in red is out of the bundle and its axis is oriented with the polar angle $\theta$ and the azimuthal angle $\varphi$ relative to the axis of the bundle. The coordinate system $(\xi_1, \xi_2, \xi_3)$ is the local coordinate system used for the flagellum outside the bundle. The radius $R$, pitch $P$ and pitch angle $\beta$ of the helical filament are also shown schematically for a red helix (not to scale).

slide along the cell to allow for different possible directions. This assumption means that the torques generated by the off-axis orientation of the filaments are neglected; as we outline in the discussion, this is reasonable given that these torques lead to additional reorientation angles that are negligible. Furthermore, this modelling approach does not account for the small section of the filament that wraps around the cell body to align it with a given axis. Here also this small section contributes to only an insignificant amount of drag and propulsion, because of the no-slip boundary condition on the cell body, and hence its impact in the overall force and torque balance can be neglected.

We use experimental observations reported in Ref. [37] to construct a detailed kinematic description. During a tumble at least one of the flagellar filaments changes their direction of rotation to clockwise (CW, see sketch in Fig 3). This in turn leads to a polymorphic transformation of that filament from the normal to the semi-coiled polymorphic form, which occurs gradually from the base of the filament to its distal end, during which the filament separates from the bundle; we denote the time for that transition to take place $t_u$. A sketch of the three polymorphic forms relevant to the model is shown in Fig 2a.

After spending a time $t_s$ in a semi-coiled form, the filament transforms to the curly 1 form, keeping its CW rotation. However, according to the data in Ref. [37], the change in the direction of swimming occurs primarily when the flagellum is in its semi-coiled form. We thus choose to neglect the transformation to the curly 1 form in the present work and focus on the normal and semi-coiled ones.

At the end of the tumble, the motor reverses back to its CCW direction and the filament transforms back to the normal polymorph while returning into the bundle, a process that takes a time $t_b$. The total change of direction resulting from the whole tumble is then measured by the angle $\gamma$, which we define as the angle between initial and final swimming directions, $\mathbf{u}_0$ and $\mathbf{u}_f$ respectively, in the lab frame.

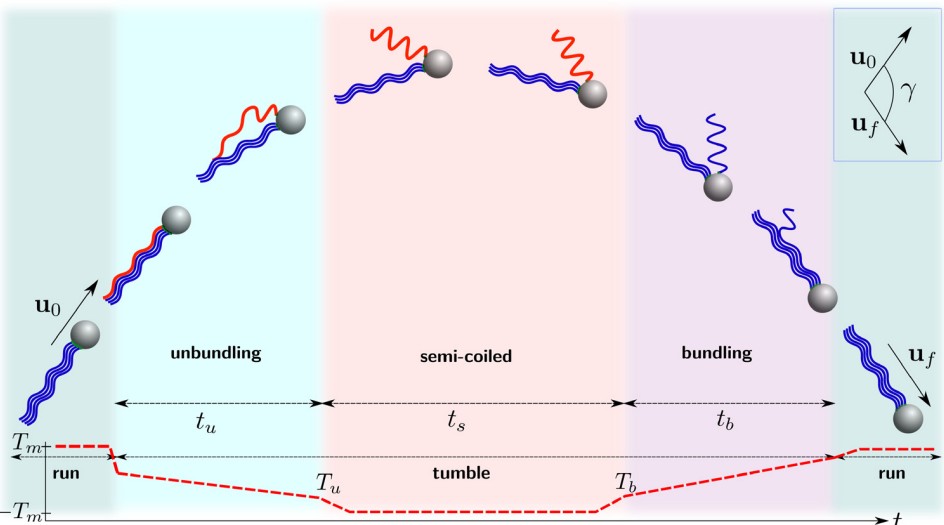

**Fig 3. Schematic representation of the run-and-tumble dynamics of a peritrichous bacterium such as *E. coli* with four flagella along different stages of tumble.** The red semi-coiled flagellum has a CW-rotating motor (when viewed from the distal cell body) while the blue normal ones rotate in the CCW direction. Bottom panel: Value of the motor torque for the filament undergoing the polymorphic transformation: (i) during runs, the torque is equal to a prescribed value $T_m$; (ii) when a tumble starts, the torque is gradually reduced, until it reaches the opposite value $-T_m$ when the flagellum has transformed to semi-coiled form; (iii) after bundling, the torque returns to the original value $T_m$. Inset: definition of the change of direction angle, $\gamma$. Figure adapted from a similar sketch in Refs. [9] and [66].

We note that the end of a tumble could be defined in two ways: either when the new swimming direction is established (this is the definition used in Ref. [38]) or when the bundle is reassembled. In this work we will use the latter definition. We assume that a tumble lasts a total time of 0.12 s, comparable with the mean value reported in Ref. [37], and that each stage of it, i.e. unbundling, semi-coiled rotation and bundling, take a third of this time, so that $t_u = t_s = t_b = 0.04$ s. As we discuss in Section 5, the precise durations of each of the three stages turn out to not play an important role in our final results.

## 2.1 Kinematic modelling of tumbling

Flagellar filaments are modelled as perfect helices with radius $R$ and pitch $P$. A left-handed helix in the local coordinate system (e.g. $(\xi_1, \xi_2, \xi_3)$ in Fig 2) is parametrised as

$$\bar{\mathbf{r}}(s) = [R\cos(2\pi s/l), -R\sin(2\pi s/l), Ps/l], \qquad 0 \le s \le L, \tag{1}$$

where $l$ is the contour length of one helical turn, i.e. $l = \sqrt{(2\pi R)^2 + P^2}$. In the case of a right-handed helix, the second term has a plus sign. The pitch angle of the helix is denoted by $\beta = \arctan(2\pi R/P)$ and we take it to be positive for a left-handed helix and negative for a right-handed one. The values of $R$, $P$ and $l$ for the different polymorphic shapes of flagellar filaments for *E. coli* are taken from Ref. [37] and are reported in Table 1.

The motor of the flagellum rotates with angular velocity $\omega$. This corresponds to an apparent helical wave so the time-dependence of the position of a flagellar filament in the frame of cell body is given by

$$\mathbf{r}(s, t) = [R\cos(2\pi s/l + \omega t), -R\sin(2\pi s/l + \omega t), Ps/l], \qquad 0 \le s \le L. \tag{2}$$

The angular velocities for the filaments leaving the bundle are chosen separately at each stage of the tumbling process. For each isolated filament in a given polymorphic form and aligned with the axis of the rotary motor (or for a bundle of filaments), the motor torque is approximately constant and of magnitude $T_m \approx 500$ pN · nm ($5 \cdot 10^{-19}$ N · m) [66]. The dominant component of the torque on the filament can be captured using resistive-force theory described in more detail in Section 2.2 and it is given by [65]

$$T = (\zeta_\parallel \sin^2\beta + \zeta_\perp \cos^2\beta)R^2 L\omega, \tag{3}$$

where the coefficients $\zeta_\parallel$ and $\zeta_\perp$ are parallel and perpendicular viscous drag coefficients

**Table 1. Geometrical parameters for the normal and semi-coiled polymorphs from Ref. [37] and the resulting propulsive forces.** The angular velocity is computed to correspond to a constant torque of $T_m = 500$ pN · nm (details in Section 2.1). The propulsive force is computed with parameters from this table using Eq (29).

| Type | Normal | Semi-coiled |
|---|---|---|
| Radius, $R$ | 0.175 $\mu$m | 0.25 $\mu$m |
| Pitch, $P$ | 2.3 $\mu$m | 1.1 $\mu$m |
| Contour length of one turn, $l$ | 2.55 $\mu$m | 1.92 $\mu$m |
| Pitch angle, $\beta$ | 25.5° | −55° |
| Angular frequency, $\omega/2\pi$ | 150 s$^{-1}$ | −94 s$^{-1}$ |
| Propulsive force, $F_{prop}$ | 0.33 pN | 0.36 pN |

respectively, approximately given by [67],

$$\zeta_{\parallel} \approx \frac{2\pi\mu}{\ln\left(L/h\right) - 1/2}, \qquad \zeta_{\perp} \approx \frac{4\pi\mu}{\ln\left(L/h\right) + 1/2}. \qquad (4)$$

We therefore find the angular frequency $\omega$ for the filament in a single polymorphic form from Eq (3) by setting $T = T_m$ and we report its values for the normal and semi-coiled polymorphic shapes in Table 1.

To model the complex elasto-hydrodynamic wrapping of the filaments around the cell body, we assume that during runs all flagellar filaments align with the $z$-axis. During tumbles, filaments are then allowed to 'slide' along the cell body to reach their prescribed locations, assuming that they are always aligned with the normal to the cell body. Physically, this sliding is enabled by the elasticity of the hook.

We now describe the kinematics of a single tumbling event as a three step process: unbundling, semi-coiled propulsion and bundling. An illustration of our modelling of tumbling for different values of the angle between the bundle and a semi-coiled filament, $\theta_0$, is provided in movies S1.avi, S2.avi and S3.avi in S1 File.

**2.1.1 Step 1: Unbundling.** During the first step of the process ('unbundling') at least one motor reverses its rotation direction to CW, so the corresponding filament separates from the bundle and transforms into a semi-coiled form, see Figs 3 and 4a. During this transition, the portion of the helix attached to the body is progressively transforming into the semi-coiled form while the distal part remains in the normal form.

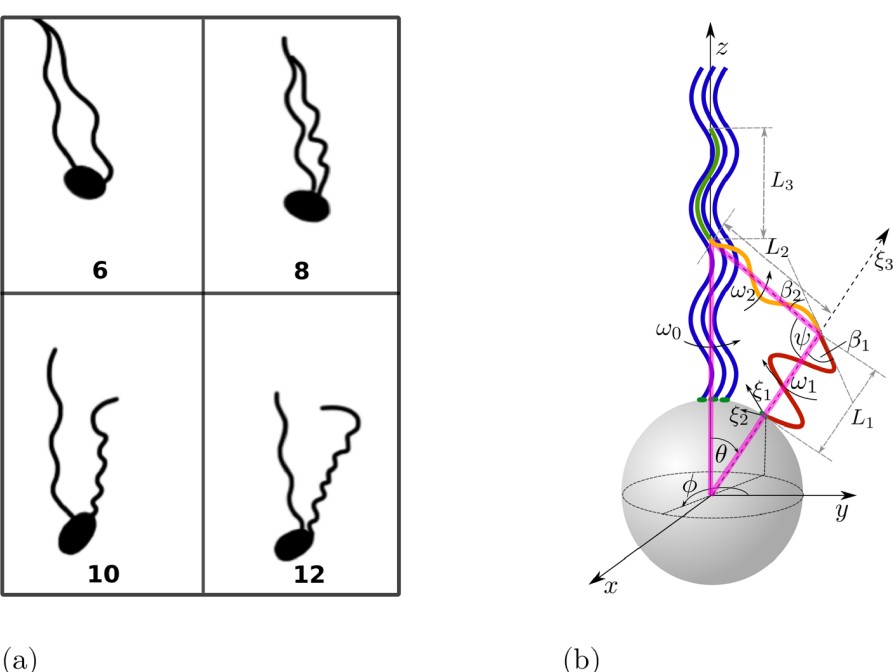

(a) (b)

**Fig 4.** (a) Illustrations of a flagellar filament undergoing polymorphic transformation and unbundling based on fluorescent images reported in Fig 7 from Ref. [37]; numbers on the drawings correspond to different frames (the time between two consequent frames is 1/60 s). (b) Schematic representation of the unbundling process. The red part of the unbundling filament is in the semi-coiled form (near the cell body), while the orange (intermediate region) and green portions (distal part) have a normal shape. The bundle consists of the other three filaments shown in blue. Note that although the kinematic model assumes that the geometric transition between the normal and semi-coiled helices is smooth, the hydrodynamic model considers only purely helical segments.

We assume that, during unbundling, the filament slides along the cell body by gradually changing the angle $\theta(t)$ that it forms with $z$-axis until it reaches a new, prescribed angle $\theta_0$. For simplicity, we take $\theta(t)$ to evolve linearly in time as

$$
\theta(t) = 
\begin{cases}
\theta_0 \dfrac{t}{t_u p}, & 0 \leq t \leq t_u p, \\[2ex]
\theta_0, & t_u p < t \leq t_u,
\end{cases}
\tag{5}
$$

where $t_u$ is the total time of unbundling and $p < 1$ is the fraction of this time during which the value of $\theta(t)$ changes. Without any further information, we make the assumption that $p$ is proportional to final angle so that $p = |\theta_0|/\pi$.

We note that our choice for the function $\theta(t)$ is somewhat arbitrary. We made an *a posteriori* check numerically using different functions $\theta$ to ensure that our results are not sensitive to this choice: we considered quadratic and square root functions for $\theta$ instead of the expression in Eq (5) and found that the difference in the total change of direction angle $\gamma$ was of the order of 1˚, which is small compared to the experimentally observed average values in the range 57˚–68˚.

During an unbundling event, the helical filament undergoes a complex polymorphic transformation. Classical studies had focused on the polymorphic transformations of the flagella of *Salmonella* [63, 64] and revealed how during this transformation, different polymorphic shapes (normal; curly; semi-coiled) are joined together. When two portions of the filament have different polymorphic shapes, the flagellum is bent and the angle between the axes of the polymorphs is given by $\psi = \pi - |\beta_1 - \beta_2|$, where $\beta_i$ ($i = 1, 2$) are the pitch angles of the two helical forms (displayed in the red and orange parts of the unbundling filament in Fig 4b). In the case of *E. coli*, based on the experimental observations in Ref. [37], we assume that the portion of the helical filament attached to the cell body is in the semi-coiled form (this is the segment $L_1$ shown in red in Fig 4) while the rest of the helical filament remains in the normal polymorph, with an angle $\psi \approx 100$˚ between helical axes (this is shown in Fig 4 in orange for the intermediate portion of the filament and in green for its distal end). We denote the region where two helices merge a 'kink'.

The motion of bistable helices have been studied theoretically in past work [68]. Two types of motion have been shown to allow propagation of kink along rotating helices. One is the crank-shafting mode, in which one helix rotates rigidly about the axis of the other. The second one is the speedometer-cable motion, where each helix rotates about its own axis with different angular velocities. Given the experimental evidence in the results from Howard Berg's group [37], we consider in our model that the kink propagation follows the speedometer-cable motion.

From the experimental images in Ref. [37], we observe that during its polymorphic transformation, a filament consists of three parts: a semi-coiled part of length $L_1$ that rotates CW with angular velocity $\omega_1$ (shown in red in Fig 4b), a normal part of length $L_2$ (orange in Fig 4b), the axis of which is inclined by an angle $\pi - \psi$, rotating with angular velocity $\omega_2$ and, finally, a part length $L_3$ which remains temporarily in the bundle (green in Fig 4b). The contour length of the first part of the filament is $L_1/\alpha_1$, with $\alpha_1 = P_1/l_1$. If $\theta < \pi - \psi$, a triangle can be formed between the bundle and the unbundling filament (shown in pink in Fig 4b). The side $L_2$ of this triangle can be estimated using the sine theorem as $L_2/\sin(\theta) = (a + L_1)/\sin(\theta + \psi)$. The third (normal) part of the filament disappears if either $L_1/\alpha_1 + L_2/\alpha_2 > L$ or if the

triangle is not formed, thus when one of the following conditions is satisfied

$$\frac{L_1}{\alpha_1} + \frac{(a + L_1)\sin(\theta)}{\alpha_2 \sin(\theta + \psi)} > L \qquad \text{or} \qquad \theta \geq \pi - \psi. \tag{6}$$

We further assume that the third (normal) part of length $L_3$ does not contribute to additional forces acting on the flagellum that unbundles, as it is still part of the bundle.

In order to model the angular velocity of the helices we follow the simplified approach proposed in Ref. [64]. We consider that the kink is moving with constant velocity $v_0$, which can be related to $\omega_1$ geometrically as

$$v_0 = -\frac{\omega_1 P_1}{2\pi}. \tag{7}$$

While the kink propagates, the contour length of the first filament is increasing at a rate $\omega_1$ $l_1/2\pi$. At the same time, the length of the second filament is decreasing at a rate $\omega_2 l_2/2\pi$. To ensure continuity at the junction point, these rates should be equal, leading therefore to the following condition relating the two angular velocities

$$\omega_1 l_1 = -\omega_2 l_2. \tag{8}$$

Eqs (7) and (8) provide us with two conditions to determine the values of $\omega_1$ and $\omega_2$. We then choose the velocity of the kink to correspond to fixed time of unbundling $t_u$, which is an experimentally observable variable; this leads to $v_0 = LP_1/2\pi l_1 t_u$. This choice of angular velocities could result in a small jump of the torque acting on the filament in between the stages of the tumble; we address this point in the following section. Note that, alternatively, one could have prescribed the value $\omega_2$ so that the exact torque $T_m$ is reached at the end of the unbundling process. This might however lead to unbundling times slightly different from those observed in the experiments.

It is important to note that deforming helices follow in general more complex dynamics during the propagation of the kink, involving the elastic energy of deformation; this, in itself, is already a complex solid mechanics problem [68, 69]. Here the model in Eqs (7) and (8), based purely on geometrical arguments, offers us a simplified approach that is analytically tractable.

**2.1.2 Step 2: Semi-coiled propulsion.** After the filament has fully unbundled, it remains in semi-coiled form during a time $t_s$ rotating in the CW direction (when viewed from the distal end), as shown in Fig 2. At $t = t_u$, the angular velocity of the filament is $\omega_1$ from the previous section, experiencing a viscous torque that is given by Eq (3), i.e. $T_u = D_{33} \omega_1$, with $D_{33} = (\zeta_{\parallel} \sin^2 \beta + \zeta_{\perp} \cos^2 \beta)R^2 L$. However, because of our modelling choice to prescribe the unbundling time, the value of this torque, $T_u$, is not necessarily exactly equal to $-T_m$. To ensure continuity, we assume that the torque changes linearly from $T_u$ to $-T_m$ over a short period of time $\delta$. In this paper we impose $\delta = t_s/10$; different choices with $\delta = t_s/20$ or $\delta = t_s/4$ affect the model predictions by about 1%. The variation of the torque in time is schematically shown in the bottom panel of Fig 3, and we can see there a linear connection between unbundle and semi-coiled stages.

A similar reasoning is applied for the torque and angular velocity immediately before the bundling stage, in order to adjust its value from $-T_m$ to $T_b = D_{33} \omega_2$ in a linear fashion during the time period between $t = t_u + t_s - \delta$ and $t = t_u + t_s$, with $\omega_2$ being the angular velocity of a semi-coiled filament in the bundling stage (with notation from Fig 5; this can also be seen in the schematic representation shown in Fig 3).

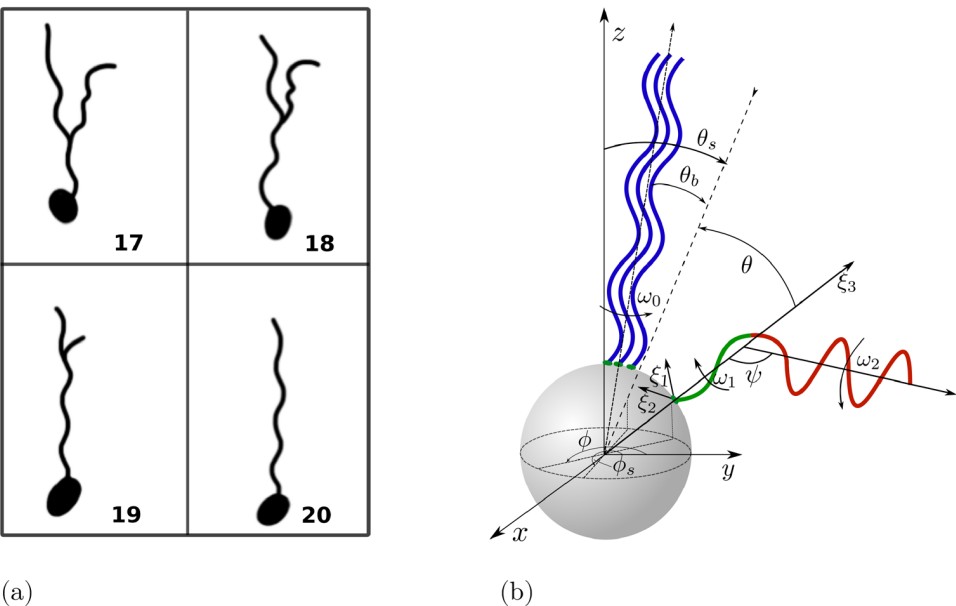

**Fig 5.** (a) Illustration of the bundling process with one filament changing its polymorphic form based on fluorescent images in Fig. 7 from Ref. [37]; numbers on the drawings correspond to different frames with a time between two consequent frames of 1/60 s. (b) Schematic representation of the bundling process. The green part of the filament is a normal polymorph (near the cell body), while the red part is in semi-coiled form (distal end). The bundle consists of the other three filaments shown in blue.

**2.1.3 Step 3: Bundling.** The third stage of the tumble is bundling of the filaments, which is modelled similarly to the unbundling process. The key difference is that instead of semi-coiled filament returning into the bundle, the bundle and the flagellum are both assumed to change their position to align with the axis of the newly-established swimming direction, thereby capturing the role played by the elastic hook joining the filaments and the cell body. This assumption is based on the empirical observations of several bundling events in Ref. [37]. These observations, along with *a posteriori* agreement with experimental values for the turning angles, justify this modelling choice.

During this zipping motion, the semi-coiled filament switches its direction of rotation back to CCW and undergoes a polymorphic transformation back to the normal form starting from its proximal end (see Fig 5a). In the model, the proximal part is a normal polymorph of length $L_1$ (shown in green in Fig 5b), while the distal part remains in the semi-coiled form and has length $L_2$ (red in Fig 5b). We assume that the filament and the bundle come together and join at the axis given by swimming direction $\mathbf{U}_s$ that was established while the flagella out of the bundle were in their semi-coiled form. This axis forms an angle $\theta_s$ with the $z$-axis, given by

$$\theta_s = \arccos\left(\mathbf{U}_0 \cdot \mathbf{U}_s / |\mathbf{U}_0||\mathbf{U}_s|\right), \tag{9}$$

where $\mathbf{U}_0 = (0, 0, -|\mathbf{U}_0|)$ was the swimming direction at $t = 0$.

In our model, we describe this transformation by simultaneously varying the values of three angles: $\theta_b(t)$, $\theta(t)$ and $\psi(t)$ (see Fig 5). Similarly to the unzipping dynamics, the angles $\theta_b$ and $\theta$

are modelled as quasilinear functions

$$\theta_b(\hat{t}) = \begin{cases} \theta_s \dfrac{\hat{t}}{t_b p}, \\[2mm] \theta_s, \end{cases} \qquad \theta(\hat{t}) = \begin{cases} \theta_0 - (\theta_0 - \theta_s)\left(\dfrac{\hat{t}}{t_b p}\right), & 0 \leq \hat{t} \leq t_b p, \\[2mm] \theta_s, & t_b p < \hat{t} \leq t_b, \end{cases} \tag{10}$$

where $t_b$ is the bundling time, $p = \max\{|\theta_0 - \theta_s|, |\theta_s|\}/\pi$ and $\hat{t} = t - t_u - t_s$. The angle $\psi$ is also assumed to change in order to preserve the continuity of the transformation, and similarly to $\theta$ it follows a quasilinear evolution

$$\psi(\hat{t}) = \begin{cases} \psi_0 \dfrac{\hat{t}}{t_b q}, & 0 \leq \hat{t} \leq t_b q, \\[2mm] \psi_0, & t_b q < \hat{t} \leq t_b, \end{cases} \tag{11}$$

where $\psi_0$ is the final angle between two helices and $q = \psi_0/\pi$. The azimuthal angles $\varphi$ and $\varphi_b$ also vary linearly during a time $t_b p$ in order to reach the value $\varphi_s$ following the path of shortest distance. We note that *a posteriori* checks allow us to verify that the specific choices for the functions $\theta(t)$, $\theta_b(t)$ and $\psi(t)$ do not have a significant effect on our final results.

We also assume that the proximal fragment of flagellum that returns to the bundle (when $\theta = \theta_s$) does not contribute to the forces and torques on the cell body, i.e. that its hydrodynamic effect is already fully captured by that of the bundle.

In this third step, the angular velocities of different parts of the filament, $\omega_1$ and $\omega_2$, are chosen in the same way as in Section 2.1.1. We note that with our modelling choice of the tumble in Fig 3, both at the beginning and at the end of the tumble the value of the motor torque for the transforming filament is not exactly equal to $T_m$ but is determined by the value of $\omega_2$ in the unbundling case and $\omega_1$ for bundling.

## 2.2 Hydrodynamic modelling of tumbling

Equipped with our geometrical model, we can now focus on the mechanical aspects of tumbling. A rigid body, which moves with linear velocity $\mathbf{U}$ and angular velocity $\boldsymbol{\Omega}$ in a fluid, produces a flow, that exerts hydrodynamic force $\mathbf{F}$ and torque $\mathbf{T}$ on the surface of this body. At zero Reynolds number the equations of motion for the fluid are the incompressible Stokes equations, which are linear. Therefore, the instantaneous force $\mathbf{F}$ and torque $\mathbf{T}$ depend linearly on $\mathbf{U}$ and $\boldsymbol{\Omega}$ via a symmetric resistance matrix

$$\begin{pmatrix} \mathbf{F} \\ \mathbf{T} \end{pmatrix} = \begin{pmatrix} \mathbf{A} & \mathbf{B} \\ \mathbf{B}^T & \mathbf{D} \end{pmatrix} \begin{pmatrix} \mathbf{U} \\ \boldsymbol{\Omega} \end{pmatrix}, \tag{12}$$

where $\mathbf{A}$, $\mathbf{B}$ and $\mathbf{D}$ are $3 \times 3$ matrices.

Bacteria swim at a Reynolds number of about $10^{-4}$, suggesting that the Stokes equations are a good approximation. For a spherical body of a bacterium, swimming with the velocity $\mathbf{U}$ and angular velocity $\boldsymbol{\Omega}$, the drag force and torque acting on it are classically given by [70]

$$\mathbf{F}_b = -6\pi\mu a \mathbf{U}, \qquad \mathbf{T}_b = -8\pi\mu a^3 \boldsymbol{\Omega}. \tag{13}$$

The cell is small so its inertia can be safely neglected [71]. As a result, it has to remain force- and torque-free, so the viscous forces and torques on flagella, denoted by $\mathbf{F}_f$ and $\mathbf{T}_f$ respectively, have to balance the viscous drag forces and torques on the cell body from Eq (13). Mechanical

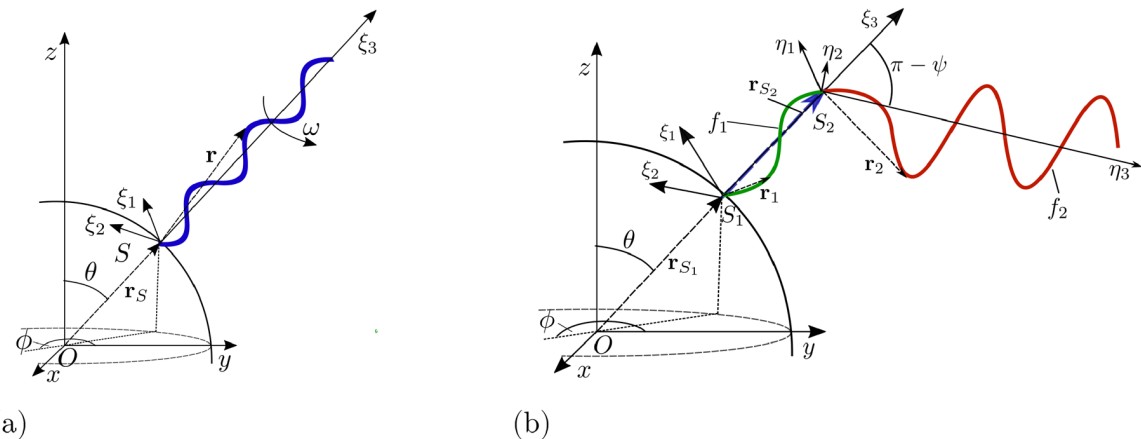

**Fig 6.** (a) Sketch of a spherical body centred at a point $O$ and a flagellar filament with a single polymorphic form attached at a point $S$ on the surface of the body. (b) Sketch of a spherical body with a filament undergoing a polymorphic transformation from shape $f_2$ (red) to shape $f_1$ (green). The point of attachment on the surface of the body is denoted by $S_1$ while $S_2$ is the junction point between the two helices.

equilibrium is therefore written as

$$\mathbf{F}_f + \mathbf{F}_b = \mathbf{0}, \qquad \mathbf{T}_f + \mathbf{T}_b = \mathbf{0}. \tag{14}$$

We now derive the expressions for $\mathbf{F}_f$ and $\mathbf{T}_f$ valid first for a single polymorphic form and then during polymorphic transformations.

**2.2.1 One polymorphic form.** Since the flagellar filaments are slender, we may use the framework of resistive-force theory [13, 65] to compute hydrodynamic forces.

Under this hydrodynamic theory, in the case of slender and weakly bent filaments (the aspect ratio of bacterial flagella is about $2 \cdot 10^{-3}$) the linear relationship in Eq (12) is in fact valid at every point along the slender filament, with a resistance matrix that depends on the relative orientation between the centreline of the filament and its velocity. Specifically, if we denote the local velocity of a flagellum by $\mathbf{u}(s, t)$, which is a function of the arclength along the helix, $s$, and time, $t$, the instantaneous hydrodynamic force density exerted on the filament is given by

$$\mathbf{f}(s, t) = -\left[\zeta_\parallel \mathbf{t}(s, t)\mathbf{t}(s, t) + \zeta_\perp \left(\mathbf{1} - \mathbf{t}(s, t)\mathbf{t}(s, t)\right)\right] \cdot \mathbf{u}(s, t), \tag{15}$$

where $\mathbf{t}(s, t)$ is the local unit tangent vector, i.e. $\mathbf{t}(s, t) = \partial \, \mathbf{r}(s, t)/\partial s$, and where the coefficients $\zeta_\parallel$ and $\zeta_\perp$ are the parallel and perpendicular drag coefficients, respectively from Eq (4).

Consider now that the flagellar filament is attached to the cell body at a point $S$, with position vector $\mathbf{r}_S$ relative to the centre of the cell body at point $O$ (see Fig 6). A typical tumble time is about 0.1 s, while the time requires for one rotation with frequency $\omega/2\pi = O(100 \text{ Hz})$ is 0.01 s, about one order of magnitude faster. It is therefore appropriate to separate these two time scales and average the flagellar forces and torques over the period of rotation, $2\pi/\omega$. The total hydrodynamic force acting along the flagellum, $\mathbf{F}_f$, can then be estimated by integrating Eq (15) along the filament and averaging it over the time period, leading to

$$\mathbf{F}_f = \frac{\omega}{2\pi} \int_0^{2\pi/\omega} \int_0^L \mathbf{f}(s, t) \mathrm{d}s \mathrm{d}t = \mathbf{A}\mathbf{U}_S + \mathbf{B}(\boldsymbol{\Omega}_S + \boldsymbol{\omega}), \tag{16}$$

where $\mathbf{U}_S$ and $\boldsymbol{\Omega}_S$ are the velocity and angular velocity of the point of attachment $S$, which can

be expressed in terms of velocities of the centre point $O$, denoted by $\mathbf{U}$ and $\boldsymbol{\Omega}$, as

$$\mathbf{U}_S = \mathbf{U} + \boldsymbol{\Omega} \times \mathbf{r}_S, \qquad \boldsymbol{\Omega}_S = \boldsymbol{\Omega}. \tag{17}$$

The angular velocity of the flagellar filament relative to the cell body, $\boldsymbol{\omega}$, is aligned with the $\xi_3$-axis, so we may write $\boldsymbol{\omega} = [0, 0, \omega]$, with magnitude $\omega$ prescribed. Similarly, the viscous torque, $\tilde{\mathbf{T}}_f$, exerted on the filament about the point $S$ is written as

$$\tilde{\mathbf{T}}_f = \frac{\omega}{2\pi} \int_0^{2\pi/\omega} \int_0^L \mathbf{r}(s,t) \times \mathbf{f}(s,t) \mathrm{d}s\mathrm{d}t = \mathbf{B}^{\mathsf{T}} \mathbf{U}_S + \mathbf{D}(\boldsymbol{\Omega}_S + \boldsymbol{\omega}), \tag{18}$$

where $\mathbf{r}$ is the vector parametrising the helical wave from Eq (2), and where we have used the representation of the forces and torques as a linear combination of velocity and angular velocity as in Eq (12). The matrices $\mathbf{A}$, $\mathbf{B}$ and $\mathbf{D}$ can be computed analytically and their values are given in S1 File. The torque on the filament evaluated at point $O$ is $\mathbf{T}_f = \tilde{\mathbf{T}}_f + \mathbf{r}_S \times \mathbf{F}_f$.

We denote by $\theta$ the polar angle between the axis of the flagellum and $z$-axis and the angle $\phi$ with $y$-axis, resulting in local coordinate system $(\xi_1, \xi_2, \xi_3)$ illustrated in Fig 6a. It is more convenient to evaluate $\mathbf{F}_f$ and $\mathbf{T}_f$ in Eqs (16) and (18) in the $(\xi_1, \xi_2, \xi_3)$ coordinates and then transform them to $(x, y, z)$ coordinates. For any vector $\mathbf{E}$ the change of coordinates rule may be written as $\mathbf{E}^{(\xi_1, \xi_2, \xi_3)} = \mathbf{M}\mathbf{E}^{(x,y,z)}$, where the matrix $\mathbf{M}$ is given by

$$\mathbf{M} = \begin{pmatrix} \cos\phi & \sin\phi & 0 \\ -\cos\theta\sin\phi & \cos\theta\cos\phi & \sin\theta \\ \sin\theta\sin\phi & -\sin\theta\cos\phi & \cos\theta \end{pmatrix}. \tag{19}$$

Using this transformation and Eq (17), the forces and torques exerted on the flagellum can be written in $(x, y, z)$ coordinates as

$$\mathbf{F}_f = \mathbf{M}^{-1}\mathbf{A}\mathbf{M}\mathbf{U} + \mathbf{M}^{-1}(\mathbf{B}\mathbf{M} - \mathbf{A}\mathbf{M}\mathbf{R}_S)\boldsymbol{\Omega} + \mathbf{M}^{-1}\mathbf{B}\boldsymbol{\omega}^\xi, \tag{20a}$$

$$\tilde{\mathbf{T}}_f = \mathbf{M}^{-1}\mathbf{B}^T\mathbf{M}\mathbf{U} + \mathbf{M}^{-1}(\mathbf{D}\mathbf{M} - \mathbf{B}^T\mathbf{M}\mathbf{R}_S)\boldsymbol{\Omega} + \mathbf{M}^{-1}\mathbf{D}\boldsymbol{\omega}^\xi, \tag{20b}$$

where the matrix $\mathbf{R}_S$ is such that $\mathbf{R}_S \mathbf{E} = \mathbf{r}_S \times \mathbf{E}$ for any vector $\mathbf{E}$. All vectors in the above expressions are given in $(x, y, z)$ coordinates, except for $\boldsymbol{\omega}^\xi$, which is expressed in the $(\xi_1, \xi_2, \xi_3)$ coordinate system for convenience. Substituting Eqs (13) and (20) into Eq (14) results in a system of linear equations for the velocity $\mathbf{U}$ and angular velocity $\boldsymbol{\Omega}$ components in the frame of the cell body.

The force balance in Eq (14) can be readily modified to account for a cell with $N > 1$ flagella, owing to linearity of the system. For the flagellum $i$ the force $\mathbf{F}_i$ and torque $\mathbf{T}_i$ can be estimated from Eq (20) with the matrix $\mathbf{M}_i$ now computed using angles $\theta_i$ and $\phi_i$. The force balance in Eq (14) is then modified to

$$\sum_{i=1}^N \mathbf{F}_i + \mathbf{F}_b = \mathbf{0}, \qquad \sum_{i=1}^N \mathbf{T}_i + \mathbf{T}_b = \mathbf{0}, \tag{21}$$

where $\mathbf{T}_i = \tilde{\mathbf{T}}_i + \mathbf{r}_i \times \mathbf{F}_i$ and where $\mathbf{r}_i$ is the vector joining the centre of the cell body to the base of the $i$th flagellum.

Using Eq (21) we can next calculate the velocity and angular velocity of the cell with $N$ flagella with different polymorphic forms, for example the situation shown in Fig 2 where the

bundle takes the form of a normal flagellum and a semi-coiled filament is separated from the bundle. However, a modification of Eq (21) is required to account for polymorphic transformations during bundling and unbundling.

**2.2.2 Two polymorphic forms.** Consider a filament that undergoes polymorphic transformation from form $f_2$ to form $f_1$ (red and green helices in Fig 6b); $f_1$ is assumed to be attached to the cell body at a point $S_1$ with position vector $r_{S_1}$ relative to the centre of the cell while the junction point between the two helices is $S_2$ at vector $r_{S_2}$. The axis of $f_2$ forms an angle $\pi - \psi$ with the axis of $f_1$. We may then introduce the local coordinate system $(\eta_1, \eta_2, \eta_3)$ by rotating $(\xi_1, \xi_2, \xi_3)$ by $\pi - \psi$ around the $\xi_1$-axis. The hydrodynamic force and torque exerted on each flagellar piece $f_i$ (with index $i = 1, 2$ corresponding to $f_1$ or $f_2$) are then given by

$$\mathbf{F}_{f_i} = \frac{\omega_i}{2\pi} \int_0^{\ell_i} \int_0^{2\pi/\omega_i} \mathbf{f}_i(s,t) \mathrm{d}t \mathrm{d}s = \mathbf{A}_i \mathbf{U}_{S_i} + \mathbf{B}_i(\mathbf{\Omega}_{S_i} + \boldsymbol{\omega}_i), \tag{22a}$$

$$\mathbf{T}_{f_i} = \frac{\omega_i}{2\pi} \int_0^{\ell_i} \int_0^{2\pi/\omega_i} \mathbf{r}_i(s,t) \times \mathbf{f}_i(s,t) \mathrm{d}t \mathrm{d}s = \mathbf{B}_i^T \mathbf{U}_{S_i} + \mathbf{D}_i(\mathbf{\Omega}_{S_i} + \boldsymbol{\omega}_i), \tag{22b}$$

where $\ell_i$ is the arclength of piece $i$ and $\omega_i$ its angular frequency (no Einstein summation is implied here). We note that the averaging over the period of rotation means that we consider perfect helices and we neglect the possible hydrodynamic effects of kinks during polymorphic transformations. The total viscous force and torque on the flagellum is then found as the sum of those for each part of the helix, and we may write

$$\mathbf{F}_f = \mathbf{F}_{f_1} + \mathbf{F}_{f_2}, \qquad \mathbf{T}_f = \mathbf{T}_{f_1} + \mathbf{T}_{f_2} + \mathbf{r}_{S_1} \times \mathbf{F}_{f_1} + \mathbf{r}_{S_2} \times \mathbf{F}_{f_2}. \tag{23}$$

To transform this result in the $(x, y, z)$ coordinate system we follow a procedure similar to the case of a single polymorphic form. For example, for the force, both $\mathbf{F}_{f_1}$ and $\mathbf{F}_{f_2}$ are written as in Eq (20a) with all the matrices having index $i$, e.g. matrix $\mathbf{A}$ is replaced with $\mathbf{A}_i$ etc. The matrix $\mathbf{M}_1$ is given by Eq (19) and matrix $\mathbf{M}_2$ is given by the product $\mathbf{M}_2 = \mathbf{M}_3 \mathbf{M}_1$ with $\mathbf{M}_3$ the rotation matrix

$$\mathbf{M}_3 = \begin{pmatrix} 1 & 0 & 0 \\ 0 & -\cos\psi & \sin\psi \\ 0 & -\sin\psi & -\cos\psi \end{pmatrix}. \tag{24}$$

Finally the angular velocities $\boldsymbol{\omega}_1$ and $\boldsymbol{\omega}_2$ are given in the $(\xi_1, \xi_2, \xi_3)$ and $(\eta_1, \eta_2, \eta_3)$ coordinate systems, respectively. The torques $\mathbf{T}_{f_1}$ and $\mathbf{T}_{f_2}$ in $(x, y, z)$ coordinates are computed analogously to the forces. At this point we can now compute the instantaneous velocity $\mathbf{U}$ and angular velocity $\mathbf{\Omega}$ at each instant of the tumble using Eq (21) where we find $\mathbf{F}_i$ and $\mathbf{T}_i$ in Eq (20) in the case of a single polymorphic form and Eq (23) when the filament is transitioning from one polymorph to another.

**2.2.3 Summary of the mathematical model of tumbles.** The steps in our model can be summarised as follows:

- Before the beginning of the tumble, $t < 0$, all flagellar filaments are bundled and aligned with the $z$-axis.

- The tumble starts at $t = 0$, with a number $N_u$ of flagella separating from the bundle (unbundling) following the kinematic laws described in Section 2.1.1.

- The values of both $\mathbf{U}$ and $\Omega$ change during this transformation and they are computed using Eqs (21)–(23) at every time step.

- At $t = t_u$ the flagella that left the bundle remain in their semi-coiled form during a time period $t_s$, as proposed in Section 2.1.2 and a swimming direction $\mathbf{U}_s$ is established.

- At $t = t_u + t_s$, the bundling process starts, following the kinematic description proposed in Section 2.1.3. The bundle and the semi-coiled flagella slide to merge along the axis set by $\mathbf{U}_s$.

- When the tumble ends at $t = t_f \equiv t_u + t_s + t_b$, the total change-of-direction angle, $\gamma$, is computed as the angle between the velocity at $t = 0$, $\mathbf{u}_0$, and that at $t = t_f$, $\mathbf{u}_f$, in the lab frame

$$\gamma = \arccos\left(\frac{\mathbf{u}_0 \cdot \mathbf{u}_f}{|\mathbf{u}_0||\mathbf{u}_f|}\right). \tag{25}$$

In order to compute velocities in the lab frame $(\hat{\mathbf{X}}, \hat{\mathbf{Y}}, \hat{\mathbf{Z}})$, which we denote by $\mathbf{u}$, we need to account for body reorientations. This is done using the change-of-coordinates matrix, $\mathbf{B}_b$, so that

$$\mathbf{u} = \mathbf{B}_b \cdot \mathbf{U}, \tag{26}$$

where columns of $\mathbf{B}_b$ consist of the unit vectors $\hat{\mathbf{x}}$, $\hat{\mathbf{y}}$ and $\hat{\mathbf{z}}$. As the cell body rotates with angular velocity $\Omega$, we advance these vectors in time as

$$\frac{d\hat{\mathbf{x}}}{dt} = \Omega \times \hat{\mathbf{x}}, \qquad \frac{d\hat{\mathbf{y}}}{dt} = \Omega \times \hat{\mathbf{y}}, \qquad \hat{\mathbf{z}} = \hat{\mathbf{x}} \times \hat{\mathbf{y}}, \tag{27}$$

which is done numerically using the function `ode45` in Matlab.

## 3 Computational results

The model outlined in Section 2 is solved using Matlab, which allows to determine numerically the time-varying linear and angular velocities of the cell body. In Fig 7 we illustrate the lab-frame trajectories of the centre of a tumbling cell equipped with two flagella; one of its flagellar filaments remains in the normal polymorphic shape throughout while the other one undergoes the change of form during the tumble. At $t < 0$, cell swims vertically in negative $\hat{Z}$ direction. During the unbundling (dash-dotted line in Fig 7), the cell gradually changes its swimming direction but undergoes only a small change of reorientation of its body frame. While the unbundled flagellum is in semi-coiled form, the cell establishes a new swimming direction in the frame of cell body, the axis of which will attract all flagellar filaments in and out of the bundle. At $t = t_u + t_s$, the bundling starts (dashed line) and the cell establishes a new swimming direction in the lab frame. The resulting change of direction, measured by the angle $\gamma$, is approximately equal to 22°, 30.4° and 47.9° for the values of the polar angle $\theta_0$ between the bundle and the semi-coiled filament of $\theta_0 = \pi/4$, $\pi/3$ and $\pi/2$, respectively.

With the description of a single tumbling event established, we now consider the statistical distribution of the angle $\gamma$ over many different tumble events. We assume that filaments are distributed uniformly on the surface of the cell body. Thus, the azimuthal angle $\phi_i$ of a flagellum $i$ has a uniform distribution on $[0, 2\pi]$ and polar angle $\theta_i$ is a random variable, such that $\theta_i = \arccos(2\nu - 1)$, with $\nu$ uniformly distributed on $[0, 1]$.

We consider cells equipped with $N$ flagella undergoing tumbling events during which a number $N_u \in \{1, 2, 3, 4\}$ of flagellar filaments unbundle and the rest remain in a normal form

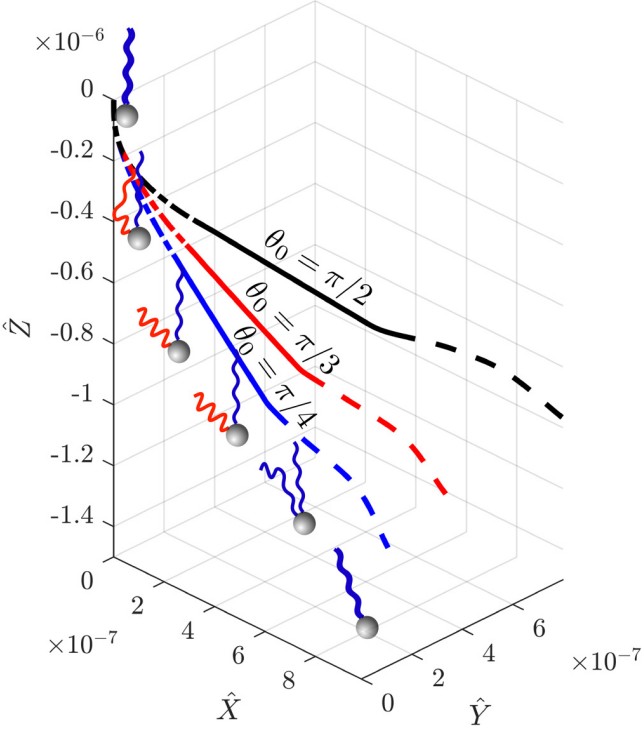

**Fig 7. Trajectory of a model bacterium with two flagellar filaments during a single tumble in which one flagellum unbundles and undergoes polymorphic transformation (normal to semi-coiled and back to normal).** The dash-dotted line represents the cell motion during unbundling; the solid line shows the dynamics of the cell when the unbundled filament is in its semi-coiled form; finally the dashed line shows the cell motion during the bundling motion. The three different trajectories are for different values of $\theta_0$ (the polar angle between the bundle and the semi-coiled filament): $\pi/4$ (blue), $\pi/3$ (red) and $\pi/2$ (black). A schematic representation of a bacterium undergoing a tumble is shown next to the blue curve for $\theta_0 = \pi/4$.

aligned with $z$-axis. We assume that the forces exerted by the tight bundle are well approximated by the forces exerted by a single filament; we address this assumption in more detail at the end of the next section. For each chosen value of $N_u$, we run the model computationally 3000 times, distributing the flagellar filaments outside the bundle uniformly on the surface of the sphere at each tumble, and computing the corresponding values of the angle $\gamma$. We then obtain the resulting probability density function (pdfs) for $\gamma$ using the Matlab function `'ksdensity'`.

We show in Fig 8 using blue solid lines the probability density function for the four different values of the number of flagella out of the bundle, $N_u$ ('Full model'). The mean change of direction for different $N_u$ is reported in Table 2. The purple circles reproduce the experimental pdf from Ref. [38] obtained using 1,166 events while the yellow triangles are the experimental data from Ref. [42] (8,058 events). We note that these experimental studies could not distinguish between different number of flagella and thus the experimental points are the same for all $N_u$ in Fig 8.

In the case where $N_u = 1$ flagellar filament undergoes a polymorphic change, the pdf produced by our model is clearly more biased towards smaller values of the angle $\gamma$ than that measured in the experiments of Refs. [38, 42]. In particular, the model leads to a maximum value of $\gamma$ of about 100°. This turns out to be in agreement with the data of Ref. [37], where the authors studied the change in swimming direction of *E. coli* and kept track of the number of filaments participating in the tumble. In their experiments, they observed that if only one

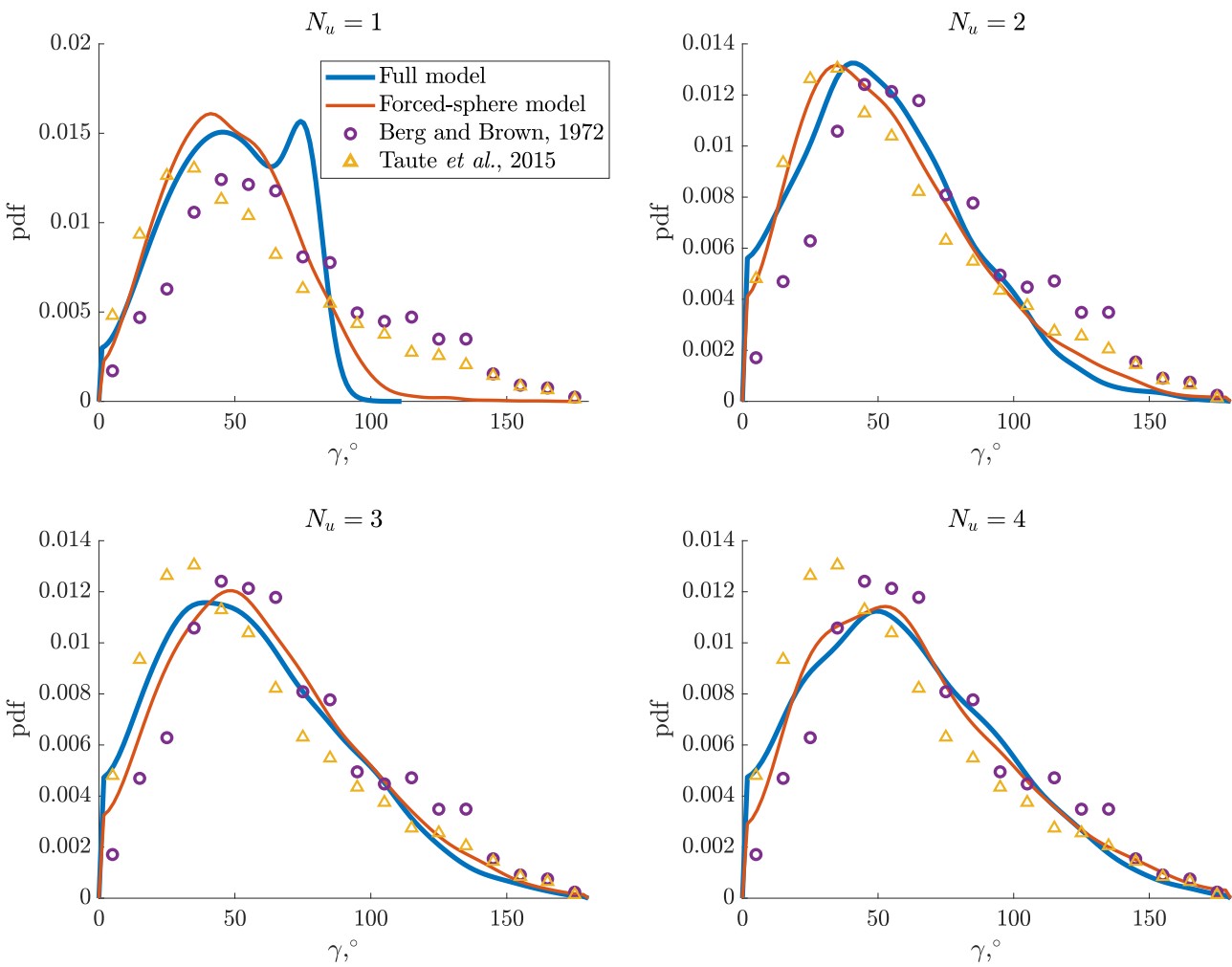

**Fig 8. Probability density functions (pdfs) for the change-of-direction angle $\gamma$ of a cell during a tumble.** For each cell, the filaments in the bundle are aligned with the $z$-axis and remains in the normal polymorphic form, while $N_u = 1$ to 4 filaments undergo polymorphic transformations during the tumble. The blue line shows the results produced by our full theoretical model described in Section 2 while the red line displays the results of the simplified forced-sphere model from Section 4. The experimental data of Berg and Brown [38] are shown purple circles while the data of Taute *et al.* [42] are shown using yellow triangles.

filament comes out of the bundle during the tumble, the change of direction is smaller, with a maximum value of about 120° in the case of a cell with two flagella. In contrast, the results of our model with $N_u \geq 2$ show a much better agreement with the experiments of Refs. [38, 42]. For $N_u = 2$, the model does give somewhat of a bias towards smaller angles, and we predict a mean reorientation angle $\langle \gamma \rangle = 52.2°$. For $N_u = 3$ and 4, we obtain $\langle \gamma \rangle = 56.7°$ and $\langle \gamma \rangle = 60°$, respectively, and the full pdf also agrees with the experimental results.

We may next use our model to determine the relative contribution to the total change of direction of the different stages of the tumble. Specifically, we can estimate numerically the reorientation of the bundle axis during the three separate stages (unbundling, semi-coiled propulsion and bundling) and we report these values in Table 2. During the first two stages, the cell reorients on average by a small angle of about 1–4° while approximately 90% of the reorientation occurs during the third stage (bundling). In that case, we may denote by $\theta_s$ the angle between $\mathbf{U}_0$, i.e. the swimming velocity at $t = 0$, and $\mathbf{U}_s$, the velocity in the semi-coiled propulsion stage, as in Eq (9). We report its mean values, denoted by $\langle \theta_s \rangle$, in Table 2, and we see that

**Table 2. Mean reorientation angle (in degrees) as a function of the number $N_u$ of flagellar filaments leaving the bundle during a tumble.**

| Mean angle (degrees) | $N_u = 1$ | $N_u = 2$ | $N_u = 3$ | $N_u = 4$ |
|---|---|---|---|---|
| **Full model $\langle \gamma \rangle$** | **47.2 ± 21.2** | **52.2 ± 30.3** | **56.7 ± 33.4** | **60 ± 34.6** |
| • Unbundling | 1.73 ± 0.65 | 1.5 ± 0.74 | 1.4 ± 0.7 | 1.3 ± 0.7 |
| • Semi-coiled propulsion | 4.1 ± 2.1 | 2.7 ± 1.7 | 2.1 ± 1.4 | 1.9 ± 1.1 |
| • Bundling | 41.7 ± 19 | 48.5 ± 29.3 | 53.8 ± 32.7 | 57.6 ± 34.3 |
| $\langle \theta_s \rangle$ | 52.34 ± 23.1 | 56.4 ± 31 | 61.3 ± 33.8 | 65 ± 35 |
| Forced-sphere model | 48.4 ± 22.7 | 55 ± 32.5 | 60.2 ± 33.8 | 63.6 ± 35.35 |
| Berg and Brown [38] | 68 ± 36 | | | |
| Turner et al. [37] | 58 ± 40 | | | |
| Taute et al. [42] | 57± 37 | | | |
| Turner et al. [43] | 60 ± 29.2 | | | |

First row: mean angle $\langle \gamma \rangle$ given by full model and during each of the three stages of the tumble (unbundling; semi-coiled propulsion; bundling). Second row: mean angle $\langle \theta_s \rangle$ between the original swimming direction and the direction of swimming during the semi-coiled propulsion (Eq (9)), and that predicted by the forced-sphere model. Third row: mean reorientation angles in the experiments of Berg and Brown [38], Turner et al. [37], Taute et al. [42] and Turner et al. [43].

they are close to the values of $\gamma$ from the full model. As with the above, this appears to suggest that the effects of unbundling and semi-coiled propulsion are subdominant compared to the bundling stage in determining the total reorientation angle for the swimming cell.

## 4 Forced-sphere hydrodynamic model

The numerical results in the previous section suggest that the tumble model can be reduced to determining the value of the angle $\theta_s$ between the pre-tumble swimming velocity and that in the semi-coiled stage. In this section, we propose therefore to simplify the tumble model further by estimating only the distribution of swimming velocities during the propulsion from the semi-coiled tumble stage.

Consider the simplest possible representation of the cell body in the semi-coiled propulsion stage of the tumble: a sphere with radius $a$ pushed by individual forces $\mathbf{F}_i$, $i = 1, \ldots, N$ representing the propulsion exerted by the propelling flagella; this is illustrated in Fig 9 in the case

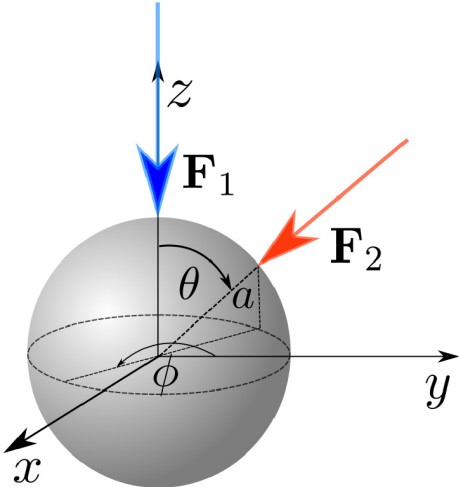

**Fig 9. Spherical cell subject to two point forces $\mathbf{F}_1$ and $\mathbf{F}_2$ representing 'phantom' propelling flagella.**

$N = 2$. We describe the location of these forces using the angles $\theta_i$ they make with the $z$-axis and their azimuthal angles $\phi_i$ relative to the $y$-axis. We align the force from the bundle (normal polymorphic shape) with the negative $z$-axis, $\mathbf{F}_1 = (0, 0, -f_1)$. The remaining forces, also directed along the normals to the cell body, are written as

$$\mathbf{F}_i = -[f_i \sin \theta_i \sin \phi_i, -f_i \sin \theta_i \cos \phi_i, f_i \cos \theta_i]. \tag{28}$$

The magnitude $f_i = |\mathbf{F}_i|$, $i = 1, \ldots, N$, is taken to be a propulsive force of the helix with radius $R_i$ and pitch angle $\beta_i$ rotating with angular velocity $\omega_i$ [65],

$$f_i = (\zeta_\perp - \zeta_\parallel) \sin \beta_i \cos \beta_i R_i L \omega_i. \tag{29}$$

Note that for normal left-handed helix $\beta_i > 0$ and $\omega_i > 0$, whereas for a semi-coiled right-handed helix $\beta_i < 0$ and $\omega_i < 0$, so that $f_i$ is always positive and the propulsive forces are always pointing towards the cell body (the values of $f_i$ for the normal and semi-coiled helices are listed in Table 1). The total force acting on the cell can therefore be written as

$$\mathbf{F}_t = -\left[ \sum_{i=2}^{n} f_i \sin \theta_i \sin \phi_i, -\sum_{i=2}^{n} f_i \sin \theta_i \cos \phi_i, f_1 + \sum_{i=2}^{n} f_i \cos \theta_i \right]. \tag{30}$$

Neglecting drag on the flagella, we can then obtain the velocity of the cell body from balancing the total propulsive force with the drag on the spherical cell,

$$\mathbf{U} = \frac{\mathbf{F}_t}{6\pi\mu a}. \tag{31}$$

In this simplified model we assume that the total change of direction during a tumble, $\gamma$, is given as the angle between negative $z$-axis, which is swimming direction at $t < 0$ (i.e. before the tumble), and $\mathbf{U}$, i.e.

$$\gamma = \arccos\left( \frac{\mathbf{U} \cdot [0, 0, -1]}{|\mathbf{U}|} \right). \tag{32}$$

For each number $N_u$ of flagellar filaments coming out of the bundle, we assume that the corresponding flagellar forces are uniformly distributed on the surface of the sphere, which we use to compute numerically the probability density functions for the change-of-direction angle $\gamma$. The results for the pdfs from this forced-sphere model are reported in Fig 8 using thin red lines and their averages are reported in Table 2 for different values of $N_u$. Although the profiles in Fig 8 are quite different for $N_u = 1$, the mean value differs only by 1.4°. For $N_u \geq 2$, the pdfs we obtain for $\gamma$ in this simplified model are very similar to those obtained by the full computational model, with less than 4° difference between the average values of $\gamma$ computed with the full and forced-sphere models. We also obtain a very nice agreement with experiments in the cases $N_u = 3$ and $N_u = 4$.

To account for the elongated shape of the cell body, we have also extended this spherical model to the case of a prolate spheroid; this is done in S1 File, where we compare the results between the model for a cell body of aspect ratio 2:1 with the forced-sphere model. We find that the results from the forced-sphere and the forced-spheroid models are in very good agreement, with mean values of $\gamma$ differing by less than 3%.

Finally, we investigate whether the number of filaments that remain in the bundle affects model predictions. Ideally this should be done by studying hydrodynamic interactions between filaments in the bundle and estimating the resulting forces. A simpler approach is

sometimes used in literature, in which the bundle with $N_b$ filaments is modelled as a single thicker filament with radius $r\sqrt{N_b}$ where $r$ is a radius of a single flagellar filament (see e.g. [61]). In our model this would lead to a change in $\zeta_\perp$ and $\zeta_\parallel$ in Eq (4), where the radius enters under the logarithm. We implemented this modification to the forced-sphere model, assuming that $N_b$ is a random integer from 1 to 5. The resulting mean angle decreases by less than 1% from the values reported in Table 2, which justifies our assumption of modelling a bundle as a single flagellum.

## 5 Summary and discussion

Research in the area of bacterial hydrodynamics has been very active over the past fifty years, but a number of fundamental questions remain unanswered. Since bacteria are present in a multitude of biological systems, it is crucial to be able to understand their role and predict their dynamics. In this paper we developed a mathematical model of bacterial tumbling to study the change of swimming direction of peritrichous bacteria. Although the organism studied in this paper is the bacterium *E. coli*, the model can be applied to study the change of direction of other peritrichous bacteria that use a run-and-tumble motility pattern. Our approach is a combination of geometrical and hydrodynamic models. We first presented a kinematic description of flagellar filaments during three stages of tumble based on fluorescent imaging performed by [37]: unbundling, semi-coiled propulsion and bundling. We then used this kinematic model to determine the linear and angular velocities of the cell body using resistive-force theory [65], which in turn allowed to solve for the change of direction angle, $\gamma$, between the swimming directions before and after the tumble. By distributing various flagella filaments uniformly on the surface of the cell body, we finally obtained the probability density function for $\gamma$. The pdf and the average value of $\gamma$ were found to be in very good agreement with the classical experiments from Refs. [38, 42], suggesting that the change of direction by swimming bacteria in a tumble is fully governed by geometry and mechanics of the interacting flagella.

From a mechanical point of view, three potential mechanisms could have been proposed, *a priori*, to govern the change of direction of a swimming bacterium. The first one is the reorientation of cell body due to the torques produced by rotation of helical filaments. However, since flagellar filaments are much longer than the cell body, when they are translating in the direction other than along their axis they experience a large drag that would resist any rotational motion. To fix ideas, in order to rotate the filament aligned with the $z$-axis along the $x$ direction (see notation in Fig 10) by an angle of $\tilde{\gamma} = 1$ rad in $\tilde{t} = 0.1$ s, which is the typical angle and duration of a tumble, one needs a torque that is larger than $T_0 = D_{11}\tilde{\gamma}/\tilde{t}$, with $D_{11}$ being first component of **D** matrix in Eq (18) (see S1 File for this expression). We may estimate this value to be about $T_0 \approx 4.9 \cdot 10^{-18}$ N · m for a flagellar filament in the normal polymorph and $T_0 \approx 1.7 \cdot 10^{-18}$ N · m for the semi-coiled one. Can torques produced by rotating filaments reach these values? Consider a cell with two point torques separated by an angle of $\pi/4$, as illustrated in case (i) from Fig 10. The magnitudes of these torques, $T_i$, $i = 1, 2$, are given by the propulsive torque on filaments rotating with angular velocities $\omega_i$, so that $T_i = D_{i,33}\,\omega_i$, with $D_{i,33}$ being a last element of $\mathbf{D}_i$ matrix for a filament $i = 1, 2$ (no summation implied). The magnitude of the torque in $x$-direction for a semi-coiled filament is $T_2 \times \sin(\pi/4) \approx 3.5 \cdot 10^{-19}$ N · m, which is about an order of magnitude smaller than $T_0$, suggesting that this mechanism is not sufficient to change swimming direction during tumble.

A second reorientation mechanism that could have been at play is the rotation of the cell induced during bundling and unbundling by the propulsive force, $\mathbf{F}_{prop}$, of an 'end piece' located at position $\mathbf{r}$ (see case (ii) in Fig 10). Since the axes of the various flagellar filaments are aligned with the normal to the sphere, the end pieces that appear during polymorphic

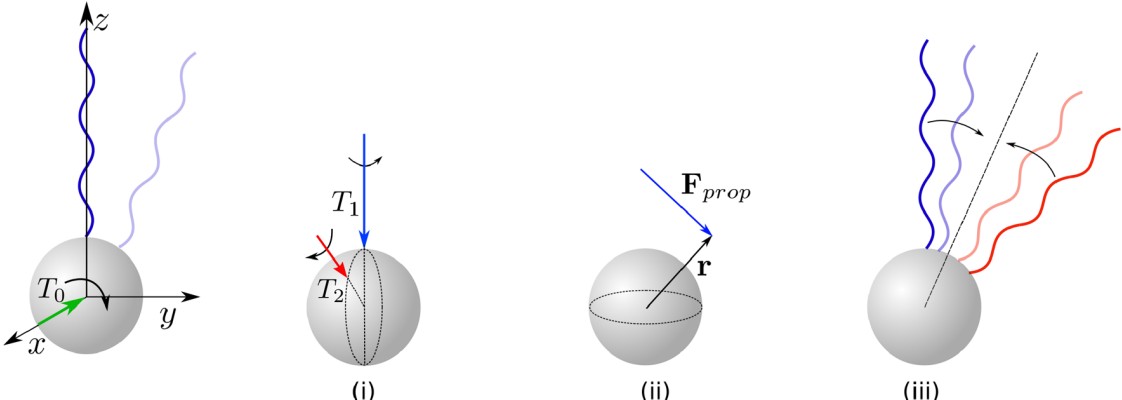

**Fig 10. A sketch of possible reorientation mechanisms.** Left: rotation of the cell body due to applied torque $T_0$ in $x$ direction; the light filament represents the position of the flagellum after the rotation. Right, case (i): a cell body subject to two point torques; case (ii): a cell body subject to a force $\mathbf{F}_{prop}$ applied at location $\mathbf{r}$ relative to the cell centre; case (iii): two flagellar filaments joining at a given axis.

transformations provide the only possible force-induced rotation. The propulsive force at every point of time can be approximated with $\mathbf{F}_{prop} \approx \mathbf{B} \cdot (0, 0, \omega)$, where $\mathbf{B}$ is the propulsion matrix from Eq (16) expressed in the local $(\eta_1, \eta_2, \eta_3)$ coordinates shown in Fig 6. For unbundling with $\theta_0 = \pi/4$, we estimate the resulting average torque to have magnitude $(\mathbf{e}_x \cdot \int_0^{t_u} \mathbf{r} \times \mathbf{F}_{prop} dt)/t_u \approx 3.2 \cdot 10^{-19}$ N $\cdot$ m, with $\mathbf{e}_x$ being a unit vector in the $x$ direction. This value is much smaller than $T_0$, suggesting that this mechanism is not responsible for cell reorientation either. Note that the estimates obtained for cases (i) and (ii) are in line with our model results: on average only a small fraction of the total change-of-direction angle $\gamma$ is due to rotation induced during the unbundling and semi-coiled stages.

The third mechanism governing the change of swimming direction is that facilitated by the elastic bending of the hook, which allows the axes of the flagellar filaments to align with the new swimming direction. The high flexibility of the hook was recently shown to be a crucial ingredient allowing 'pusher' cells (i.e. organisms pushed from the back by their filaments, such as *E. coli*) to swim in the first place [50]. In this work we assumed that the axes of the filaments are always aligned in the direction normal to the sphere while the flexibility of the hook allows the filaments to slide along the cell body, as schematically shown in Fig 10(iii). Using this model, we then obtained that the majority of the change of direction for the cell occurs during bundling, which suggests it as the main reorientation mechanism. We further note that the first two mechanisms (cases (i) and (ii) in Fig 10) rely on the duration of the tumble, whereas the experiments of Ref. [42] suggest that the modulation in turning angle does not come from differences in tumble time. In contrast, the third mechanism (iii) is not affected by tumble duration.

To further validate this understanding of the reorientation mechanism, we proposed a simplified model that consists of a sphere forced by propulsive forces, representing the forcing from 'phantom' flagellar filaments. The swimming direction in this case is aligned with the total force acting on the cell body. The resulting model is fully analytical, is able to reproduce experimental results and is in a very good agreement with the full model.

One aspect of cell motion that we neglected is the role of rotational diffusion for the cell. The rotational diffusion coefficient $\mathcal{D}_r$ of a filament in the direction orthogonal to its axis is given by the Stokes-Einstein formula, $\mathcal{D}_r = kT/D_{11}$, where $k$ is Boltzmann's constant, $T$ is the absolute temperature and $D_{11}$ is the first element of the resistance matrix $\mathbf{D}$ in Eq (18).

Assuming for simplicity one-dimensional angular motion, the mean square angular displacement is then given by $< (\Delta\gamma)^2 > = 2\mathcal{D}_r\Delta t$. With $T$ = 293 K and using parameters from Section 2, we find $\mathcal{D}_r \approx 8.3 \cdot 10^{-3}$ rad$^2$/s for a filament in the normal polymorphic shape. For a tumble duration $\Delta t$ = 0.1 s, this then leads to a contribution of rotational diffusion to the reorientation of about 2.3˚, and thus negligible.

The full model made several additional assumptions that are worth mentioning. Firstly, we model the cell body as a sphere, when the actual cell body for a swimming bacterium is closer to a spheroid with aspect ratio of about two. Several numerical works have previously modelled cells as spheroids [50, 61] or capsules [72]. Extending the full model to a spheroid would require modifications in both the geometrical and kinematic description of the present model. The forced-sphere model is, itself, easily modified to the case of a forced-spheroid, assuming that the axes of the flagellar filaments always pass through the centre of the body and that they are uniformly distributed on the surface of the cell. This is done in S1 File where we find results very similar to the forced-sphere case, with mean values of $\gamma$ differing at most by 3%.

We also assume in our model that the duration of each stage of the tumble is the identical (a third of the total duration of a tumble), which could be further refined based on numerical and experimental studies. However, as long as the total tumble duration is of the order of 0.1 s, as seen experimentally, body reorientation due to torques (cases (i) and (ii) from Fig 10) will remain insignificant in comparison to effects of the flexibility of the hook (case (iii)), which itself does not depend on time but is purely geometric.

Finally, in our work we do not model explicitly the elasticity of the hook but only implicitly include its effect through the sliding of the flagella along the cell body during bundling. It would be possible, although significantly more complex, to explicitly include this effect by combining our model with a fluid-structure solver [50]. Hydrodynamic interactions between the cell body and the flagellar filaments, and between multiple filaments, could also be included, and resistive-force theory itself could be replaced with slender-body theory to improve accuracy of the model predictions. However, both would require the use of computational solvers for the fluid flows and hydrodynamic forces, and as such would be much more involved to implement than the model proposed in the current study. The simplicity of the model in our paper will allow, in future work, to vary the various modelling parameters and thus to easily test further hypotheses about the geometry and hydrodynamics of tumbling bacteria.

## Supporting information

**S1 File. Contains the expression for resistance matrix, namely matrices A, B and D from section 2.2 and the description of forced-spheroid model.** The movies `S1.avi`, `S2.avi` and `S3.avi` are for the kinematic model of the tumble with $\theta_0 = \pi/4$, $\theta_0 = \pi/3$ and $\theta_0 = \pi/2$, respectively.
(ZIP)

## Acknowledgments

The authors thank T. S. Shimizu and K. M. Taute for kindly providing raw data for the turning angles. We would also like to thank M. Tătulea-Codrean and D. Das for useful suggestions on this manuscript and to C. Esparza-López for valuable discussions on helical dynamics.

## Author Contributions

**Conceptualization:** Mariia Dvoriashyna, Eric Lauga.

**Formal analysis:** Mariia Dvoriashyna.

**Funding acquisition:** Eric Lauga.

**Investigation:** Mariia Dvoriashyna, Eric Lauga.

**Methodology:** Mariia Dvoriashyna, Eric Lauga.

**Project administration:** Eric Lauga.

**Resources:** Eric Lauga.

**Software:** Mariia Dvoriashyna.

**Supervision:** Eric Lauga.

**Validation:** Mariia Dvoriashyna.

**Visualization:** Mariia Dvoriashyna.

**Writing – original draft:** Mariia Dvoriashyna, Eric Lauga.

**Writing – review & editing:** Mariia Dvoriashyna, Eric Lauga.

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
