## [Decision Letter · Decision Letter 0]

6 May 2021

PONE-D-21-10002

Hydrodynamics and direction change of tumbling bacteria

PLOS ONE

Dear Dr. Dvoriashyna,

Thank you for submitting your manuscript to PLOS ONE. After careful consideration, we feel that it has merit but does not fully meet PLOS ONE’s publication criteria as it currently stands. Therefore, we invite you to submit a revised version of the manuscript that addresses the points raised during the review process.

We look forward to receiving your revised manuscript.

Kind regards,

Fang-Bao Tian

Academic Editor

PLOS ONE

Additional Editor Comments:

Dear Mariia and Eric,

Thank you for submitting your work to the Plos One. It has been reviewed by three referees in the field. All agree that this is a nice work, but needs to be revised carefully. I will look forward to receiving the revised version.

Regards,

Fangbao

Journal Requirements:

2. Please amend your list of authors on the manuscript to ensure that each author is linked to an affiliation. Authors’ affiliations should reflect the institution where the work was done (if authors moved subsequently, you can also list the new affiliation stating “current affiliation:….” as necessary).

Reviewers' comments:

Reviewer's Responses to Questions

**Comments to the Author**

1. Is the manuscript technically sound, and do the data support the conclusions?

Reviewer #1: Yes

Reviewer #2: Yes

Reviewer #3: Partly

2. Has the statistical analysis been performed appropriately and rigorously? 

Reviewer #1: Yes

Reviewer #2: N/A

Reviewer #3: Yes

3. Have the authors made all data underlying the findings in their manuscript fully available?

Reviewer #1: Yes

Reviewer #2: No

Reviewer #3: Yes

4. Is the manuscript presented in an intelligible fashion and written in standard English?

Reviewer #1: Yes

Reviewer #2: Yes

Reviewer #3: Yes

5. Review Comments to the Author

Reviewer #1: This is a great piece of work that models the bundling and unbundling process of e.coi. Of course, I also find that there are so many assumptions adopted, which might affect the readability of the work. Here are a few minor points:

1, hight-handed -> right-handed, line 144.

2, what is the difference between r in eq 1 and bar{r} in eq 2?

3, line 162, i can understand that sliding can be enabled by elastic hooks, however, when this happens, it will be difficult to think that the filaments are still normal to the cell body? any supporting information for this?

4, in fig. 4, the definition of \\psi can be difficult to understand, according to its definition on line 190.

5, eq 6, could the authors provide more details on how this is derived? to me, this may not look that intuitive.

Reviewer #2: In general, this paper is interesting.

But there are some issues to be addressed.

1. The authors intended to model E.coli which is a rod-shape bacteria. Why the authors use a spherical particle to represent the rod-shape bacteria? In recent years, the following paper has carefully addressed the modelling of Rod-shape bacteria in fluid flow.

Modelling bacterial twitching in fluid flows: a CFD-DEM approach, Scientific Reports, 2019, 648915

1. The manuscript is wordy and seems like a chapter of PhD thesis. It should be condensed.

2. It would be better to clarify the new physical insights we could learn from the modelling, which cannot be obtained by experiments alone.

Reviewer #3: This article studies the hydrodynamics of "run-and-tumble" motion of E. coli and other bacteria with multiple flagella. First, a detailed model is presented in which various stages of a tumble (unbundling, semi-coiled propulsion, and bundling) are described kinematically and incorporated into a hydrodynamic model. The authors identify the stage that contributes most to reorientation and use the results to construct a "minimal" model that reproduces similar statistical distributions of reorientation angles.

Overall, I think the results and conclusions from these models are valid and the manuscript is clearly written. Statistics are obtained with a fairly large sample size. Relevant literature is succinctly summarized to give context and motivation for the model.

The main shortcoming of the methodology is that there are a lot of ad hoc assumptions about the behaviour of the flagella in place of a mechanistic model. This is understandable for the sake of obtaining a simple conceptual model but raises questions about consistency and physicality.

For example, is it valid to allow the point of connection between the cell body and flagellum to slide around the cell? There should be a section of the flagellum that wraps around the body from the motor to the bundle and this should contribute to the force and torque balance equations. (Please see below for related remarks.)

Also, it appears from the model that the changing orientation of the bundle is largely responsible for the reorientation. Is there experimental evidence that the orientation of the bundle relative to the cell changes as in the model? If so, it would be surprising if the non-spherical shape of E. coli cells were not important. Presumably, there are preferred orientations, altering the distribution of random angle changes.

Below are some other specific comments and questions.

1. The comparison with experiments focused on distributions of reorientation angle. Does the trajectory as a whole match experimental observations, e.g., during the semi-coiled propulsion phase?

2. The figures depicting the model flagella are slightly misleading. The model considers pure helical segments, which do not connect to the cell body or to each other as a smooth curve. It should be noted that the representations in the figures are not the actual geometrical model of the filament.

3. It is not clear why the kink velocity should be related to the rotation rate as in Eq. 7. It should be more complicated than this, e.g., dependent on the energy density of the two helical forms.

4. In line 232 and a few other locations, there are remarks like "T_u is not necessarily -T_m". If the torque should be -T_m, why not choose omega_1 and omega_2 so that this is true? Is this essentially what it means when "the torque changes linearly from T_u to -T_m"? Or is Eq. 3 being modified over time? Even though it is checked that the outcome is not sensitive to the choice of delta, I struggle to see why this step is necessary.

5. Is it physically justified to compute the torque on the flagella using Eq. 3 and setting T_m = T? The motor torque should be applied in a fixed direction (e.g., normal to the cell body), whereas the axes of the flagella are changing throughout the unbundling/bundling process and, moreover, the bundle moves so the total motor torque is different before and after tumbling. Is this possible if the motors are fixed on the cell body?

6. In Eq. 22, should the omega at the start of the LHS be omega_i? The segment lengths l_i change with time but my interpretation is that time averaging is done for a fixed l_i at each instant in time. Is this correct?

6. PLOS authors have the option to publish the peer review history of their article (what does this mean?). If published, this will include your full peer review and any attached files.

Reviewer #1: No

Reviewer #2: No

Reviewer #3: No

---

## [Author Response · Author response to Decision Letter 0]

25 Jun 2021

Responses to reviewers are in 'PloS ONE - answers to 1st round reviews.pdf' file, which was attached earlier along with modified manuscript.

---

## [Editor Report · Decision Letter 1]

29 Jun 2021

Hydrodynamics and direction change of tumbling bacteria

PONE-D-21-10002R1

Dear Dr. Dvoriashyna,

We’re pleased to inform you that your manuscript has been judged scientifically suitable for publication and will be formally accepted for publication once it meets all outstanding technical requirements.

Kind regards,

Fang-Bao Tian

Academic Editor

PLOS ONE
---

## [Editor Report · Acceptance letter]

8 Jul 2021

PONE-D-21-10002R1 

Hydrodynamics and direction change of tumbling bacteria 

Dear Dr. Dvoriashyna:

I'm pleased to inform you that your manuscript has been deemed suitable for publication in PLOS ONE. Congratulations! Your manuscript is now with our production department. 

Kind regards, 

on behalf of

Dr. Fang-Bao Tian 

Academic Editor

PLOS ONE